# Deciphering Interactions between Phosphorus Status and Toxic Metal Exposure in Plants and Rhizospheres to Improve Crops Reared on Acid Soil

**DOI:** 10.3390/cells12030441

**Published:** 2023-01-29

**Authors:** Xiurong Wang, Shaoying Ai, Hong Liao

**Affiliations:** 1Root Biology Center, State Key Laboratory for Conservation and Utilization of Subtropical Agro-Bioresources, South China Agricultural University, Guangzhou 510642, China; 2Institute of Agricultural Resources and Environment, Guangdong Academy of Agricultural Sciences, Guangzhou 510640, China; 3Root Biology Center, Fujian Agriculture and Forestry University, Fuzhou 350002, China

**Keywords:** acid soil, phosphorus, aluminum, manganese, cadmium, interaction, phosphate signaling, resistance, crop improvement

## Abstract

Acid soils are characterized by deficiencies in essential nutrient elements, oftentimes phosphorus (P), along with toxicities of metal elements, such as aluminum (Al), manganese (Mn), and cadmium (Cd), each of which significantly limits crop production. In recent years, impressive progress has been made in revealing mechanisms underlying tolerance to high concentrations of Al, Mn, and Cd. Phosphorus is an essential nutrient element that can alleviate exposure to potentially toxic levels of Al, Mn, and Cd. In this review, recent advances in elucidating the genes responsible for the uptake, translocation, and redistribution of Al, Mn, and Cd in plants are first summarized, as are descriptions of the mechanisms conferring resistance to these toxicities. Then, literature highlights information on interactions of P nutrition with Al, Mn, and Cd toxicities, particularly possible mechanisms driving P alleviation of these toxicities, along with potential applications for crop improvement on acid soils. The roles of plant phosphate (Pi) signaling and associated gene regulatory networks relevant for coping with Al, Mn, and Cd toxicities, are also discussed. To develop varieties adapted to acid soils, future work needs to further decipher involved signaling pathways and key regulatory elements, including roles fulfilled by intracellular Pi signaling. The development of new strategies for remediation of acid soils should integrate the mechanisms of these interactions between limiting factors in acid soils.

## 1. Introduction

Solubilized aluminum (Al) destroys plant membranes, inhibits nutrient uptake, impairs photosynthesis, and eventually hampers plant root and shoot growth, as does the toxic element cadmium (Cd), and high concentrations of the essential nutrient element manganese (Mn) [1,2,3]. Simultaneously, metal stress frequently induces oxidative stress [4,5,6]. Many plants species are capable of alleviating symptoms of toxicities caused by Al and other heavy metals through the actions of phosphorus (P), an essential nutrient for plant growth and development [7,8,9]. As a central player in plant metabolism, plant phosphate (Pi) signaling networks are modulated by a variety of environmental signals, including the availability of other elements [10,11]. In order to fully utilize P as mediator of resistance to Al, Mn, and Cd toxicities in crop improvement efforts, it is crucial to understand the intricate regulatory mechanisms by which plants modulate toxicity responses and improve P use efficiency.

Though P may be prevalent in a field, most of it is typically inaccessible for plant growth due to adsorption to soil particles, tight chemical bonding such as oxides and hydroxides of iron (Fe) and Al, and the formation of insoluble complexes with calcium. Only a very small fraction of total P is readily available in soil solution as inorganic orthophosphate P (H_2_PO_4_^−^) [12,13,14]. To mobilize P from Pi-limited soils, plants have evolved a variety of adaptations, including the development of root morphological and architectural traits that are advantageous for P acquisition, the enhanced root exudation of protons, organic anions and phosphatases, interactions with P-solubilizing microorganisms, and the activation of high-affinity Pi transporters [15,16]. Phosphate transporter (PHT) family proteins play a role in Pi uptake from soil and Pi translocation within different plant organs [17]. In roots, Pi must be transported against a steep extracellular–intracellular concentration gradient to enter plants, which is mainly achieved by PHT1 located in the plasma membranes of root cells [18].

Acid soils often cooccur with deficiencies in essential nutrient elements, typically P, potassium, or calcium, as well as, toxicities of metal elements, such as Al, Mn, Fe, and Cd [1,19]. As the most abundant metal on earth, Al toxicity has long been recognized as a major limiting factor of crop production on acid soils (pH < 5) [20]. The initial and most dramatic effect of Al toxicity is the inhibition of root growth and activity [21]. Manganese toxicity is another major factor limiting plant growth in acid soils. The symptoms of Mn toxicity are mainly the formation of chlorotic and necrotic leaf spots, accompanied by the inhibition of photosynthesis and the accumulation of excessive reactive oxygen species (ROS) [22]. The heavy metal Cd is extremely toxic to plants in acid soils, where low pH substantially increases Cd bioavailability, acquisition by plant roots, and translocation to aerial parts [23]. Excess Cd causes a number of toxic symptoms in plants, with stunting and chlorosis common. Additionally, Cd toxicity can induce efflux of cations and the generation of free radicals [24].

Recent molecular, biochemical, morphological, and physiological reports contribute tremendously to understanding plant P nutrition more clearly and comprehensively. However, mechanisms on P alleviation of toxicities of Al, Mn, and the heavy metal Cd, and potential applications for crop improvement, has been scarcely explored. This review paper mainly assesses the literature on the role of P in alleviating toxicities of all these three toxic metals. We also review recent advances in the role of P sensing and associated gene regulatory networks to cope with Al, Mn, and Cd toxicities. These discussions provide new insights into interactions between P and toxic metals, which can be utilized for crop improvement.

## 2. Phosphorus Nutrition and Al Toxicity Stress

### 2.1. Uptake, Transport and Distribution of Al in Plants

In root systems, the first site for Al uptake is the root apex, where Al toxicity inhibits root cell expansion and elongation, along with later cell division [1,20]. Cell walls, plasma membranes and cytoplasms are the main targets of Al injury, and also play important roles in Al toxicity response [20,25]. Cell walls serve as the first barrier to Al uptake at the root apex. More than 80% of total accumulated Al in plant roots is tightly bound to cell walls, with only a small fraction entering the cytoplasm, where much of that is inactivated through chelation or sequestration in the vacuoles of both roots and leaves for Al tolerance [26,27]. The binding of Al to cell walls decreases extensibility and cell elongation, and also increases rigidity [28,29]. In one documented example, Al is removed from rice cell walls and sequestered into vacuoles via concerted activity of plasma membrane–localized Nramp (natural resistance-associated macrophage protein) Al transporter 1 (Nrat1) and tonoplast-localized ATP-binding cassette (ABC) transporter Al sensitive 1 (ALS1) products [30,31].

### 2.2. Aluminum Resistance in Plants

Plants have developed two types of strategies to cope with Al toxicity. One is external detoxification that prevents Al from entering the root apex, and the other is internal detoxification through compartmentalization of Al into vacuoles and root cell walls [20,32].

The most well-documented external Al tolerance mechanism involves the Al-induced secretion of organic acids (mainly oxalate, citrate and malate) from root apices, which can chelate and detoxify Al in the rhizosphere [33]. In fact, the first report of an Al resistance gene came in revealing Al-induced efflux of malate from wheat root tips through aluminum-activated malate transporter 1 (ALMT1) products [34]. Since then, the Al-induced citrate transporter multidrug and toxic compound extrusion/Al-activated citrate transporter 1 (MATE/AACT1) has been successively identified in both monocots and dicots [35,36,37,38,39].

Aluminum entering plant roots faces the first line of internal detoxification in cell walls. The major cell wall components pectin and hemicellulose are themselves capable of binding Al [27,40]. Under Al stress, plants may also increase resistance by reducing the production of xyloglucans through the down-regulation of the xyloglucan endotransglucosylase/hydrolase gene *XTH31* [41]. Plants attempt to sequester free Al into vacuoles through transporters identified in rice and Arabidopsis as tonoplast-localized ALS1 transporters [31,42].

Al toxicity induces the expression of many Al resistance genes by directly activating transcription factors, such as sensitive to proton rhizotoxicity 1 (STOP1), Al resistance transcription factor 1 (ART1), abscisic acid, stress and ripening 5 (ASR5), and WRKY22. AtSTOP1 and OsART1 are two related members of C_2_H_2_-type zinc-finger transcription factor family that exert similar Al resistance functions in both Arabidopsis and rice [20]. In Arabidopsis, AtSTOP1 positively regulates the expression of numerous Al-induced genes, including *AtALS3*, *AtALMT1*, *AtMATE*, *AtGDH1* and *AtGDH2* (glutamate-dehydrogenase 1 and 2), which are known to participate in Al tolerance [43,44]. In rice, OsART1 regulates the expression of 31 downstream genes involved in both internal and external detoxification of Al. These include OsSTAR1 and OsSTAR2 (sensitive to Al rhizotoxicity 1 and 2) for modification of cell walls, OsNrat1 for transport of Al^3+^ into root cells and subsequent sequestration into vacuoles by OsALS1, MATE citrate transporter OsFRDL4 for secretion of citrate, magnesium (Mg) transporter OsMGT1 for increase of Mg uptake, and cysteine-rich peptide OsCDT3 to inhibit Al ingress into root cells. The expression of *OsFRDL4* is also regulated by ASR5 and WRKY22 [33,45,46,47]. 

### 2.3. Phosphorus–Aluminum Interactions in Soils and Plants

Aluminum toxicity and P deficiency often coexist in acid soils, with P fixation in Al-P precipitates being a significant contributor to low P availability [48]. Aluminum toxicity further inhibits root development and reduces the ability of plants to acquire P from soils, which ultimately reduces crop yield. Aluminum stress may also directly interfere with plant P metabolism [49]. One area of overlap is that both P deficiency and Al toxicity can independently induce synthesis and the secretion of organic acids, which are important for P solubilization and Al chelation [50]. 

The influence of P nutrition on Al detoxification is still under debate, with interesting effects observed at both ends of P availability. Under P deficiency conditions, decreased Al binding to cell walls and plasma membranes can ameliorate the severity of Al toxicity as organic acid exudation increases, and as membrane phospholipids are replaced with non-Pi galactolipids and sulfolipids [7,51]. This is consistent with results from tomato, in which P deficiency reduces Al toxicity by inducing the exudation of phosphoenolpyruvate carboxylase-derived organic acids [52]. On the other hand, ample P supplies are known to alleviate Al toxicity in many crops [53]. In sufficient availability, P can enhance Al tolerance not only through the direct immobilization and detoxification of Al by P in the rhizosphere and root tissues, but also through indirect interactions associated with improved root development and nutrient uptake, as well as through the stimulated exudation of Al-chelating organic acids [7,54].

Observations of variation among genotypes has revealed that the effects of P on Al tolerance are largely associated with P efficiency and Al resistance. Along those lines, the positive effects of P application may be more significant in lines exhibiting more resistance to Al toxicity [7]. In one example, P enhances Al resistance in Al-resistant *Lespedeza bicolor* but not in Al-sensitive *L. cuneata* when grown for 30 days with alternate Al and P treatments in a hydroponic system [55]. Similarly, Al-tolerant cultivars may utilize P more effectively than Al sensitive cultivars [53,56]. The over-expression of Al-resistance gene *TaALMT1* increases P uptake in transgenic barley in comparisons with non-transformed sibling lines growing on acid soils in short-term pot trials, but there are no significant differences between the genotypes when the soil is limed. [57]. Moreover, P-efficient genotypes can be more Al tolerant than related P-inefficient genotypes in both *Stylosanthes* and soybean in solution culture [48,56]. This might be attributable to more effective P acquisition with P-efficient genotypes resulting from the enhanced exudation of organic acids from roots exposed to Al, and thusly protecting roots from Al toxicity [48,58]. 

## 3. Phosphorus Nutrition and Mn Toxicity Stress

### 3.1. Uptake, Transport and Redistribution of Mn in Plants 

Manganese (Mn) is mainly present in soils as Mn^2+^, Mn^3+^, and Mn^4+^. Of these, divalent Mn^2+^ is the most soluble and available form for plant acquisition, though the concentrations of each form in soil solution are affected by soil pH and redox conditions [59]. Bioactive Mn^2+^ accumulation is generally observed at soil pH values below 5.0, or in poorly drained soils [60]. Unlike Al, Mn uptake occurs in mature root regions, from which it is transported to root tips via phloem vessels, or translocated to shoots through xylem connections [61]. In Arabidopsis, root Mn uptake is mainly mediated through the plasma membrane-localized transporter Nramp1, with some contribution from the ZIP (zinc-regulated transporter/iron-regulated transporter-like protein) transporter IRT1 [60]. In rice, the polarly localized transporters OsNramp5 and OsMTP9 (metal tolerance protein 9) participate in Mn uptake and influx from the rhizosphere towards the stele through directed actions on opposite sides of the cell [62,63]. In barley, HvNramp5 is a transporter that is constitutively required for Mn uptake, while HvIRT1 participates in Mn uptake only under Mn-deficient conditions [64].

Most acquired Mn is translocated to shoots, with excess Mn being stored in vacuoles to regulate cellular Mn homeostasis. Two Arabidopsis ZIP family members, AtZIP1 and AtZIP2, are involved in Mn translocation from roots to shoots [65]. In rice, the yellow stripe-like protein OsYSL2 is responsible for long-distance transport and distribution of Mn [66]. At the basal nodes of rice, OsNRAMP3 functions in Mn distribution from the xylem to the phloem, and various shoot tissues [67]. The tonoplast-localized Mn transporter ShMTP8 in *Stylosanthes hamata* was the first characterized transporter for Mn sequestration into vacuoles [68]. AtMTP8 from Arabidopsis and OsMTP8.1 from rice are also localized to the tonoplast and responsible for Mn tolerance [69,70]. Additionally, specific cation exchanger (CAX) family proteins have also been implicated in Mn efflux into vacuoles, as among them AtCAX2 from Arabidopsis and OsCAX3 from rice [71,72].

### 3.2. Manganese Resistance in Plants

Plants can use several strategies to cope with excess Mn, including the regulation of Mn acquisition and translocation, chelation and compartmentalization of Mn in subcellular sites (e.g., apoplasts, vacuoles, endoplasmic reticuli (ER), and Golgi bodies), and the activation of antioxidant systems [61,73,74].

Plants maintain Mn homeostasis through activities of several transporters representing a diverse set of metal transporter families. The acquisition of Mn occurs via influx from the rhizosphere, while a several transporters move it internally across the cytosol into various organelles or into the apoplast and cell walls. Products implicated in Mn translocation and sequestration include NRAMPs, MTPs, ZIPs, YSLs, and CAXs, as mentioned above. In one notable example, AtMTP11 in Arabidopsis localizes to Golgi-like compartments participating in the vesicular trafficking and exocytosis of excess Mn, while its rice homologue OsMTP11 sequesters Mn into trans-Golgi compartments that are transported into vacuoles [75,76]. Two other Arabidopsis ER-type Calcium ATPase family members, AtECA1 and AtECA3, also contribute to Mn tolerance by exporting excess Mn from the cytosol into ER and Golgi bodies, respectively [77,78].

The synthesis and exudation of organic acids is also an important strategy plants employ to alleviate Mn toxicity. Mn can be complexed by organic acids, such as oxalate, citrate and malate, in the plant or rhizosphere, thereby increasing plant tolerance to excess Mn [79,80]. In *Stylosanthes guianensis*, superior Mn tolerance is achieved by the coordination of internal and external Mn detoxification through malate synthesis and exudation, which is regulated by SgMDH1 accumulation at both transcription and protein levels [81].

Antioxidant systems, including antioxidant enzymes and nonenzymatic components, also play important roles in alleviating ROS oxidative stress triggered by Mn toxicity. Antioxidant enzymes, such as superoxide dismutase, peroxidase, catalase, ascorbate peroxidase, glutathione reductase, and nonenzymatic antioxidant components, such as ascorbate and glutathione, are effective ROS scavengers that participate in Mn tolerance in many plant species [82,83,84,85]. As yet, signaling and key regulatory mechanisms underlying Mn activation of antioxidant systems remain largely unknown.

### 3.3. Phosphorus–Manganese Interactions in Soils and Plants

Excess Mn in soils modifies the absorption, translocation, and utilization of other mineral nutrients, such as P, calcium, Fe, thereby causing injury to plants [73]. Conversely, some nutrient elements can ameliorate Mn toxicity effects on plants. Notably, interactions between P and Mn are observable in both soils and plants [86,87]. In hydroponic systems, interactions between P and Mn occur mainly in roots during Mn uptake. By contrast, plants grown in acid soils with pH 5.1 exhibited lower leaf Mn concentrations in response to the elevation of P supplies [87]. Similar contributions of P towards alleviating Mn toxicity impacts are also found in soybean (*G. max*), potato (*Solanum tuberosum*), perennial ryegrass (*Lolium perenne*) and white clover (*Trifolium repens*) [88,89,90]. Mechanisms backing this alleviation appear to involve the formation of insoluble P and Mn complexes in soils and plants, along with P supplementation reducing oxidative stress and improving chloroplast ultrastructure [91,92].

Under P deficiency conditions, exudation of carboxylates into the rhizosphere can increase P availability by mobilizing sparingly soluble mineral P, and thus increase plant P acquisition [15]. Meanwhile, the release of carboxylates can also mobilize a range of micronutrients including Mn. Therefore, some plant species that release relatively large amounts of carboxylates to improve P acquisition also tend as a side effect to accumulate relatively high levels of Mn in leaf tissues [87,93]. In contrast, under P sufficiency conditions, the absence of this carboxylate-releasing P-acquisition strategy may reduce Mn acquisition. Phytate, as a dominant form of soil organic P, also has the potential to chelate Mn and other divalent cations [60]. Elevated P levels may lead to a reduction in phytase exudation from roots, which reduces the catalytic hydrolysis of phytate, and thereby may decrease Mn release from phytates and uptake into plants in calcareous soils [94]. It indicates that P supply has also potential to reduce Mn availability in acid soils.

Since the release of carboxylates from plant roots may increase the availabilities of both P and Mn, high leaf Mn concentrations can be used as a valuable screening parameter to screen for high P acquisition efficiency in crop breeding programs [93]. However, high P supplies might only alleviate Mn toxicity in the Mn-sensitive genotypes grown under hydroponic conditions subjected to increased P and excess Mn treatments, as in ryegrass [9]. Conversely, varieties with differences in P acquisition capacities do not necessarily lead to variety-specific differences in Mn sensitivity in a hydroponic culture with alternate P and Mn treatments [95]. In short, relationships between P efficiency, P availability and resistance to Mn toxicity remain unclear and require further study.

## 4. Phosphorus and Cd Toxicity Stress

### 4.1. Uptake, Transport and Redistribution of Cd by Plants

Cadmium exists in soil solution predominantly as divalent Cd^2+^, which is the most toxic form of Cd. The accumulation of Cd in plants is mediated by several transport processes, including uptake by roots, xylem-mediated long-distance transport from roots to shoots, phloem redistribution into seeds, and sequestration into vacuoles of roots and shoots [24]. The movement of Cd to plant roots mainly depends on transpiration-driven mass flow of the soil solution [96]. Cadmium uptake by roots is carried out by metal transporters for essential or beneficial elements, such as Fe (divalent cations), Mn, and Zn. In Arabidopsis, the root uptake of Cd is mediated mainly by the Fe transporter AtIRT1 and the Mn transporter AtNRAMP1 [97,98]. In rice, the Mn transporter OsNramp5 is a chief transporter for root uptake of Cd [62]. Additionally, Cd can also enter rice plant root cells through the Mn transporter OsNRAMP1, the major facilitator family protein OsCd1, the Fe transporters OsIRT1 and OsIRT2, and the Zn transporters OsZIP9 and OsZIP5 [99,100,101,102,103].

After roots uptake Cd, it may be translocated to shoots, which is a key process in controlling Cd accumulation in shoot tissue. Heavy metal P1B-type ATPase (HMA) is responsible for root-to-shoot translocation in both monocots and dicots, with known examples from Arabidopsis, rice and barley including AtHMA2 and AtHMA4, OsHMA2, and HvHMA2 [104,105,106]. In rice, OsZIP7 and the defensin like protein OsCAL1 also play roles in Cd translocation from roots to shoots [107,108].

Shoot Cd are distributed among different tissues and organs by several key transporters. OsLCT1 (low-affinity Cd transporter 1) is a plasma membrane transporter and plays a role in intervascular transfer, which transports Cd from enlarged vascular bundles connected to lower nodes or leaves to diffuse vascular bundles connected to upper nodes or panicles in rice nodes. Knockout of *OsLCT1* may result in rice plants with less Cd in phloem and grains [109]. OsHMA2 and OsZIP7 are detectable in the plasma membrane of phloem companion cells in both enlarged and diffuse vascular bundles, and also mediate the intervascular transfer of Cd in rice nodes toward the grain [108,110].

Resistance to Cd toxicity typically involves sequestration in vacuoles. Cadmium is transported into vacuoles by AtHMA3 in Arabidopsis and OsHMA3 in rice, and is thereby largely restricted from further translocation [111,112,113]. OsABCC9 is another Cd vacuolar transporter that appears to mediate the sequestration of Cd into vacuoles of rice roots, with knockouts of *OsABCC9* leading to increased Cd accumulation in rice grains [114]. Once in vacuoles, Cd is largely sequestered. However, there may be routes of efflux, as overexpression of *NRAMP3* and *NRAMP4* in Arabidopsis increases sensitivity to Cd [115].

### 4.2. Cadmium Resistance in Plants

To cope with Cd toxicity, plants have evolved many defense mechanisms that include decreasing Cd uptake and accumulation, sequestrating Cd into metabolically inactive parts such as vacuoles and cell walls, synthesis of phytochelatins and metallothioneins to bind Cd, and increasing activities of enzymatic and nonenzymatic antioxidants to alleviate resulting stress [24,116].

As presented above, a variety of transporters participate in Cd uptake, translocation, and distribution in plants [117,118]. Altering expression levels of these transporters might be a way to limit Cd accumulation in edible parts of crops. For example, the overexpression of *OsHMA3* in rice leads to increased sequestration of Cd into root vacuoles, limited Cd translocation to shoots, and strong reductions in grain Cd accumulation, all whilst having little effect on plant growth or on accumulation of other essential micronutrients in the grain [112,119,120]. However, the manipulation of transporters to manage Cd accumulation does not always come free of side effects. In another example involving rice, knockout of *OsNRAMP5* using CRISPR-Cas9 gene editing technology may not only reduce Cd accumulation in roots, shoots, and grains, but also causes a reduction of Mn uptake and rice growth in low Mn soils [62,121,122].

Cell walls provide a natural barrier to prevent Cd entry. Cadmium can be stored in cell walls through binding with negatively charged ions on hemicelluloses and pectins, including carboxyl and hydroxyl groups [6,123]. In rice, the increase of pectin and hemicellulose contents is closely associated with increased Cd retention in cell walls [124].

Plants can also tolerate Cd stress by synthesizing metal chelates and complexes. Cadmium is known to bind to phytochelatins (PCs), glutathione (GSH) and metallothioneins (MTs), which can be useful for minimizing the concentration of free Cd in different plant parts. Both PCs and GSH chelate Cd to facilitate Cd sequestration from cytosols into vacuoles [96,118]. Plus, simultaneously enhancing PC and GSH biosynthesis can effectively increase Cd tolerance in Arabidopsis [125].

Similar to impacts of Mn toxicity, the antioxidant system is also an important defense strategy used by plants to deal with Cd toxicity. The activation of the antioxidant system can scavenge excessive ROS, and thereby preserve redox equilibrium in plant cells, and keep plant membrane functionally sound [24].

### 4.3. Phosphorus–Cadmium Interaction in Soils and Plants

Interaction between P and Cd have been noted in both soils and plants. Although the overuse of mineral P fertilizers can lead to excessive accumulation of Cd in agricultural soils, only 2.2% of the Cd applied with P fertilizers reaches plant shoots [126,127]. Bioavailability of Cd in soils is controlled by many factors, including pH, redox potential, prevalence of organic matter, and concentrations of nutrient elements [128]. The application of P fertilizers, for one, not only supplies crops with available Pi, but also directly promotes the formation of insoluble Cd-Pi complexes, as well as, increases the negative surface charge of soils to enhance Cd adsorption to soil constituents, including clays, metal oxides and hydroxides, carbonates, phosphates, and soil organic matter [128,129,130,131]. Additionally, the application of some P fertilizers, such as calcium magnesium phosphates and phosphate rocks, increases soil pH and decreases Cd uptake by plants [128,132].

After Cd enters into plants, P can limit the mobility of Cd through binding Cd to the cell wall fraction [133,134]. In rice, applying P promotes the distribution of Cd to iron plaques and cell walls of root tissues [128,132]. Increasing P application can also increase the contents of organic acids in root tissues, which enhances Cd accumulation in the roots and decreases translocation to shoot tissues [8]. Applied P itself may directly restrict Cd translocation from roots to edible plant parts by locking up significant portions of Cd within plants into Pi-Cd complexes [135]. Consistent with the idea of Cd inactivation in complexes with P, a significant positive correlation between Cd and P has been observed in different parts of many crops [8,133,136,137]. In addition, P application may alleviate Cd-induced oxidative damage by decreasing malonaldehyde content, increasing proline content, enhancing the activities of antioxidant enzymes, and inducing the synthesis of non-protein thiols, GSH and PCs [134,138].

Conversely, Cd toxicity also has an effect on plant P nutrition. Elevated Cd concentrations decrease plant P contents in *Pinus sylvestris* planted in sand culture with pH 4.5 [139]. In soybean, high Cd concentrations may inhibit plant P uptake in P-efficient genotypes but not in P-inefficient genotypes in sand culture with pH 5.5 [140]. In contrast, Cd stress promotes P accumulation in rice when grown in a hydroponic system with different P and Cd treatments [141]. Specific interactions between P and Cd might depend on Cd tolerance strategies employed by plants. Along those lines, Cd stress has been observed to increase shoot P concentrations in a Cd hyperaccumulating population, but not in a Cd excluding population of Cd tolerant *Arabidopsis helleri* when grown in solution culture with different Cd treatments [142]. Given these divergent results among research involving plants employing a variety of Cd tolerance responses, further study is needed to elucidate the complex interaction between P and Cd in different plant species and genotypes.

## 5. Phosphate Signaling and Its Roles in Alleviating Toxicity of Al, Mn, and Cd

### 5.1. Phosphate Signaling in Plants

To adapt to low Pi availability, plants have activated local and systemic signaling pathways to sense external and internal Pi concentrations and enhance Pi acquisition and allocation [143]. Phosphate is a primary signaling molecule that triggers the generation of various secondary signals from hormones, sugars, P-containing metabolites, peptides, mobile RNAs and ROS, all of which participate in a multitude of local responses to Pi availability, with alterations in root morphology, architecture, and soil exploration perhaps most prominent [143]. Numerous molecular actors participate in plant responses to Pi deprivation conditions. Of note here, phosphate deficiency response 2 (PDR2), low phosphate root 1/2 (LPR1/2) and STOP1-ALMT1 are all Arabidopsis products known to participate in local Pi sensing and regulation of primary root growth [144,145]. In addition, root growth factor (RGF) peptides and hormones co-ordinate to promote root hair and lateral root growth in response to low P conditions [146].

Beyond local Pi signaling pathways, plant P uptake is controlled by systemic Pi signaling. Phosphate starvation response (PHR) transcription factors, such as AtPHR1 in Arabidopsis and OsPHR2 in rice, are the central regulators of this systemic Pi signaling. AtPHR1/OsPHR2 activates expression of a large set of low-Pi responsive genes by binding to PHR1 binding site (P1BS) elements [147,148,149]. For example, AtPHR1 promotes increased Pi uptake by directly inducing expression of *PHT1*, as well as of *PHOSPHATE TRANSPORTER TRAFFIC FACILITATOR 1* (*PHF1*), which facilitates PHT1 egress from the ER to the plasma membrane under low Pi conditions [150].

A group of proteins containing the SYG1/Pho81/XPR1 (SPX) domain have been identified as key sensors and regulators of Pi homeostasis and signaling through the ability to negatively regulate the binding affinity of PHRs to P1BS elements in manner that depends on Pi availability [151,152]. The binding of SPX1/SPX2 proteins to AtPHR1 in Arabidopsis, as well as OsPHR2 in rice, occurs under Pi-sufficient conditions, which prevents binding to P1BS elements and the downstream promotion of many low-Pi responsive genes. Conversely, under Pi-limited conditions, SPX1/SPX2 is degraded via the 26S proteosome pathway, and thusly frees PHR1 to bind to P1BS elements and thereby activates downstream Pi starvation-induced gene expression [148,151,152].

### 5.2. The Role of Pi Signaling in Plant Resistance to Al, Mn, Cd

In comparison with our deep understanding of Pi status signaling studied in isolation, much less is known about how Pi signaling might be involved in plant resistance to Al, Mn, and Cd toxicities. The transcription factor STOP1 is activated by multiple edaphic factors, including low Pi availability, low pH, and Al toxicity. PHR-independent activation of STOP1 can increase the expression of malate efflux transporter *ALMT1*, which is required for transcriptional regulation of low-Pi responses at root tips and root morphological changes under low Pi conditions [144,145]. Therefore, organic acid exudation may not only protect roots from Al toxicity, but also redesign root development to explore soil more likely to be favorable for P acquisition [145]. Similarly, ALS3/STAR1 is involved in the inhibition of primary root growth under Pi limitation, and is also involved in Al tolerance [153,154]. Recent work also shows that P addition can increase the levels of the signaling molecules IAA and NO, and simultaneously inhibit the expression of several heavy metal transporters, thus lowering Cd content and increasing Cd resistance in rice [155]. Further work to test this P contribution to Cd resistance is required throughout plant life cycles, as work with rice seedlings no significant connection between external Pi availability or internal Pi-starvation signaling with Cd accumulation [141].

In plants, Al, Mn, and Cd toxicities result in the production of ROS, which can be associated not only with cell damage but also with plant signaling responses [156]. As secondary signals of primary Pi signals, ROS are involved in regulation of key root developmental responses to Pi starvation [143,145]. The generation of ROS can trigger local root responses to both Pi deficiency and Al toxicity, which often coexist in acid soils and may converge on shared processes [157]. Reactive oxygen species can also directly regulate defensive responses to heavy metals or changes in cellular redox balance [158]. In sorghum, ROS production seems necessary for root citrate exudation, the induction of *SbMATE* gene expression, and the Al tolerance of root cells [35,159]. In addition, PHR1 is potentially involved in an ROS burst necessary for root tip inhibition through direct targeting of the SPX-containing vacuolar transporter chaperone 4 (VTC4) complex [145].

Various secondary signals, including sugars, hormones (e.g., auxin, ethylene, strigolactone and gibberellin), peptides, phloem mobile RNAs, and P-containing metabolites are also involved in both provoking diverse Pi starvation responses and participating in Al, Mn, and Cd tolerance [20,23,143,160]. Further studies are necessary to fully understand the networks involved in plant responses to P deficiency and heavy metal stress, particularly those common signals regulate cellular responses to multiple stresses.

## 6. Applying Knowledge of P Interactions with Al, Mn, and Cd in Crop Improvement Efforts on Acid Soils

Acid soils are characterized by high Al and Mn concentrations, as well as low P availability that limit crop production. Meanwhile, the heavy metal Cd is extremely toxic to plants in acid soils. In past decades, various agricultural practices have been employed to manage Al, Mn, and Cd toxicities. The in situ immobilization of Al, Mn, and Cd is considered one of the most effective techniques to lower the bioavailability of these elements. The application of Pi fertilizers not only may relieve P deficiency symptoms, but can also ameliorate toxicities of Al, Mn, and Cd in acid soils. In addition to reacting directly with metal ions in both soils and plants, applied P also alters soil characteristics, such as pH, Pi availability and surface charge, each of which contributes to reducing the mobility and bioavailability of metal ions for plant uptake from soils and translocation in plants [7,74,128,161]. In paddy soils, flooding induces reducing conditions in the rhizosphere region, thereby increasing P availability in acid soils due to the reduction of ferric Pi to more soluble ferrous form, which can cause Pi induced adsorption and precipitation of the metals [128].

Overall, tremendous progress has been realized in elucidating molecular mechanisms driving toxic element transport and translocation, and many genes involved in toxic element uptake and detoxification pathways have been identified. This knowledge may be important in efforts to engineer crops exhibiting reductions in toxic element accumulation relative to sibling and predecessor plants. Exploring variants in the release of organic acids promises to be an effective strategy for developing crops that are adapted to acid soils based on simultaneous protection against P deficiency and toxicities of Al, Mn, and Cd [8,50,81]. In such cases, genetic improvements in plant resilience to toxic mineral exposure might be achieved through increases in organic acid synthesis and/or exudation. For example, over-expression of the malate transporter *TaALMT1* involved in Al tolerance can promote P acquisition for transgenic barley plants grown on acidic soils in short-term pot trials, but there are no significant differences between the genotypes when the soil is limed. [58]. Plus, transgenic canola (*Brassica napus*) lines overexpressing a *Pseudomonas aeruginosa* citrate synthase gene display increases in citrate synthesis and exudation and thusly carry improved tolerance to Al toxicity and P deficiency under P-deficient conditions or exposure to Al stress, and also possess improved P uptake in soils with pH 6.19 and FePO_4_ used as the sole P source [162]. Intriguing as these results might be, the effectiveness of these genes for crop improvement in real field conditions remains far from certain. Even less certain has been progress towards finding key genes encoding organic anion transporters capable of simultaneously enhancing P efficiency and tolerance to Mn and Cd toxicity. Though relationships between P nutrition and detoxification of Mn and Cd have been experimentally established through common organic acid pathways, significant work remains to identify all key actors and important targets for crop improvement efforts.

## 7. Conclusions

Based on the studies carried out in hydroponic systems and on several acid soils, the role of P in enhancing plant resistance to metal toxicities (Al and heavy metals) has been linked to its immobilization of these toxic elements in the rhizosphere, cell walls and cytosols, as well as its regulation of organic acid biosynthesis and the exudation of organic acid anions, such as citrate and malate. Possible roles of Pi in alleviation of Al, Mn, and Cd toxicities through signaling pathways, and in soils and plants, are summarized in Figure 1. The development of new strategies for the remediation of acid soils should integrate the mechanisms of these interactions between limiting factors in acid soils. Further work is needed to sufficiently understand the interacting transcriptional, translational, and post-translational mechanisms underlying plant adaptations to acid soils, particularly the associated deficiencies of P and toxicities of Al, Mn, and Cd, for breeding programs to most effectively produce crops adapted to acid soils. Among possible projects, an understanding of the roles played by intracellular Pi signaling in alleviating toxicities of Al, Mn, and Cd remains sparse and needs to be further elucidated.

## Figures and Tables

**Figure 1 cells-12-00441-f001:**
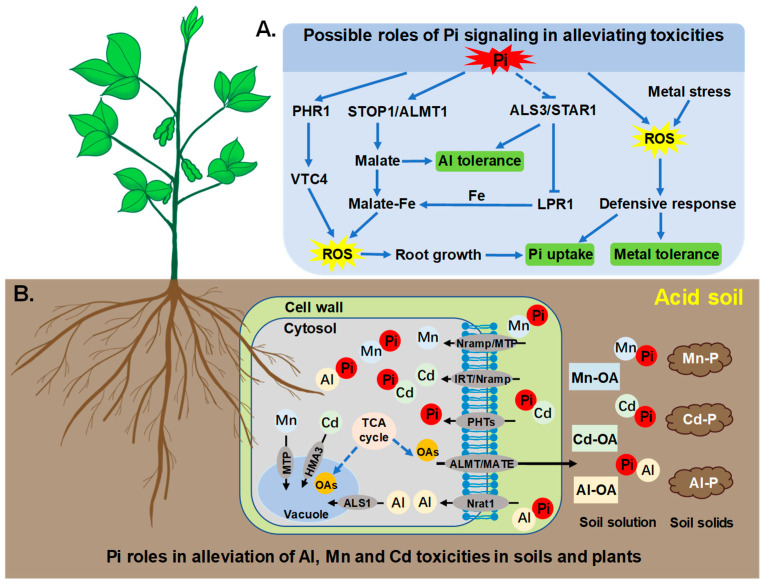
Schematic model explaining Pi roles in alleviation of Al, Mn, and Cd toxicities through signaling pathways (**A**), and in soils and plants (**B**). (**A**) ALS3/STAR1, STOP1/ALMT1, and LPR1 interactions regulate root growth by generating ROS, and thereby promote Pi uptake under Pi deficiency conditions. ALS3/STAR1 and STOP1/ALMT1 modules also participate in Al tolerance. PHR1 is potentially involved in ROS bursts necessary for inhibition of root tips through direct targeting VTC4 complex. Reactive oxygen species can act not only as secondary signals to the primary Pi signal, but also signal metal stress themselves to directly regulate plant responses directed towards enhancing resistance to both Pi deficiency and metal toxicities. Arrows indicate promotion, and perpendicular lines indicate suppression. Dotted lines indicate indirect interactions. (**B**) Phosphate application can reduce uptake and translocation of Al, Mn, and Cd to shoots due to possibly forming complexes of Pi-Al, Pi-Mn, and Pi-Cd on soil solids, or in soil solution, root cell walls, and cell cytosols. Another possibility is that Pi may enhance synthesis of organic acids to chelate Al, Mn, and Cd, which leads to less soluble Al, Mn and Cd, and declines in toxic effects on plant cells and metabolism. Nramp, natural resistance-associated macrophage protein; Nrat1, Nramp aluminum transporter 1; MTP, metal tolerance protein; ALS1/3, Al sensitive 1/3; IRT, iron-regulated transporter-like proteins; ALMT, Al-activated malate transporter; MATE, multidrug and toxic compound extrusion; OAs, organic acids; PHT, Pi transporter; TCA, tricarboxylic acid; PHR1, Phosphate starvation response 1; HMA3, Heavy metal P1B-type ATPase 3; STOP1, sensitive to proton rhizotoxicity 1; STAR1, sensitive to Al rhizotoxicity 1; LPR1, low phosphate root 1; VTC4, SPX-containing vacuolar transporter chaperone 4 complex; ROS, Reactive oxygen species.

## Data Availability

No new data were created or analyzed in this study. Data sharing does not apply to this article.

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
