# Peer review of "Deciphering Interactions between Phosphorus Status and Toxic Metal Exposure in Plants and Rhizospheres to Improve Crops Reared on Acid Soil"

_cells, 2023, doi:10.3390/cells12030441_

Round 1

Reviewer 1 Report

General comments

Reviewing the mitigation or alleviation of acid soil Al, Mn and Cd toxicities with phosphorus is more of an original and rigorous study and may be considered for publication in cells once reviewer comments are addressed. 

It would be very useful to present the importance of this study in terms of the expanse and extent of acid soils particularly cultivated. Rather, the authors have briefly mentioned that acid soils account for approx. 50% of the world's arable soils (the book reference in turn is not underpinned by some original geographic study). So, the authors do not provide any acid soil data on cultivated soils to showcase the importance of this review. It is unexpected to not find any review on the extent and expanse of acid soils. Yes, the authors.

No data on study designs or statistical designs are provided for the acid soils reference studies. The review studies are expected to present the comprehensive details of the studies included. 

Many references are from neutral, slightly alkaline and/or alkaline soil-plant studies - and this is the drawback of this study and should be rectified. 

The figure and conclusion are informative but could provide some take-home messages to suggest policy development - here and at the end of the abstract. 

Author Response

Responses to the editor and reviewers (Original comments/questions in italics followed by our answers/responses):

Reviewer #1:

Reviewing the mitigation or alleviation of acid soil Al, Mn and Cd toxicities with phosphorus is more of an original and rigorous study and may be considered for publication in cells once reviewer comments are addressed.

Answer: Thank you for your encouragements and suggestions. We revised the text according to your suggestions. Please see the following response, and revised manuscript for the details.

It would be very useful to present the importance of this study in terms of the expanse and extent of acid soils particularly cultivated. Rather, the authors have briefly mentioned that acid soils account for approx. 50% of the world's arable soils (the book reference in turn is not underpinned by some original geographic study). So, the authors do not provide any acid soil data on cultivated soils to showcase the importance of this review. It is unexpected to not find any review on the extent and expanse of acid soils. Yes, the authors.

Answer: Thanks for your concerns and suggestions. We checked the literature and also asked some experts in soil science to try to know the expanse and extent of acid soils particularly cultivated in the world. It is pity that no relevant statistic data is reported except that one original paper published in 1995 by von Uexküll and Mutert in Plant and Soil mentioned that as much as 50% of the world’s potentially arable lands were acidic.

No data on study designs or statistical designs are provided for the acid soils reference studies. The review studies are expected to present the comprehensive details of the studies included.

Answer: Thanks for your concerns and suggestions. We added some description of study designs or statistical designs for the acid soil reference studies according to your suggestions. Please see the revised manuscript for the details.

Many references are from neutral, slightly alkaline and/or alkaline soil-plant studies - and this is the drawback of this study and should be rectified.

Answer: Thanks for your concerns and suggestions. The studies on interactions of P nutrition with Al, Mn, and Cd toxicities including possible mechanisms driving P alleviation of these toxicities were often carried out in hydroponic experiments, where the solutions were adjusted to pH 4.2 for Al treatments, or 5.0-5.5 for P, Mn and Cd treatments. We checked up all references to ensure correct references. For the references about the potential applications of P nutrition for crop improvement on acid soils, we added some description of study designs according to your suggestions. Meanwhile, we also rectified some mistakes in references. Please see the revised manuscript for the details.

The figure and conclusion are informative but could provide some take-home messages to suggest policy development - here and at the end of the abstract.

Answer: Thanks for your concerns and suggestions. We added a take-home message to suggest policy development in Abstract and Conclusion sections according to your suggestions. Please see the following responses, and Abstract and Conclusion sections of the revised manuscript for the details.

Development of new strategies for remediation of acid soils should integrate the mechanisms of these interactions between limiting factors in acid soils.

Reviewer 2 Report

I greatly enjoyed reading this paper; it is a very useful review of the molecular interactions between phosphate and heavy metal toxicity to plants growing on acid soil. The paper convincingly shows, on good biochemical evidence, that under phosphate deficiency, plants will have more trouble resisting toxic metal stress. The paper also illustrates the biochemical basis for phosphorous fertilization of crops in acid soils.

The manuscript is very well written; I cannot find any flaws. The addition of a nice summarizing figure greatly adds to the attractiveness of the paper.

While the paper can be published essentially as it is, I suggest a few issues for consideration by the authors:

-       The insights derive mostly from work on rice and Arabidopsis. These are not known to be particularly acid-tolerant plant species. Would the inclusion of work on species with a natural tolerance against acidity and aluminum toxicity help in this review?

-       Phosphate mobility is known to depend greatly on soil redox status and iron speciation. The reduction of iron hydroxide and ferri-phosphate under anaerobic conditions, e.g. in rice culture, will promote the release of phosphate into the soil solution. Any reference to iron speciation and redox state is missing in this paper. Is it not relevant?

-       Phosphate is also known to interact with lead (Pb). Precipitation with phosphate is a well-known mechanism for detoxifying lead in both plants and animals. Is there any relationship with the matter discussed in this paper?

Author Response

Responses to the editor and reviewers (Original comments/questions in italics followed by our answers/responses):

Reviewer #2:

I greatly enjoyed reading this paper; it is a very useful review of the molecular interactions between phosphate and heavy metal toxicity to plants growing on acid soil. The paper convincingly shows, on good biochemical evidence, that under phosphate deficiency, plants will have more trouble resisting toxic metal stress. The paper also illustrates the biochemical basis for phosphorous fertilization of crops in acid soils.

The manuscript is very well written; I cannot find any flaws. The addition of a nice summarizing figure greatly adds to the attractiveness of the paper.

Answer: Thank you for your encouragements and suggestions. We revised the text according to your suggestions. Please see the following response, and revised manuscript for the details.

While the paper can be published essentially as it is, I suggest a few issues for consideration by the authors:

- The insights derive mostly from work on rice and Arabidopsis. These are not known to be particularly acid-tolerant plant species. Would the inclusion of work on species with a natural tolerance against acidity and aluminum toxicity help in this review?

Answer: Thanks for your suggestions. You are right. The inclusion of work on species with a natural tolerance against acidity and aluminum toxicity will help in this review. However, the impressive amount of progress in Al resistance mechanisms has been made mainly in the model plants of monocots and dicots, such as rice and Arabidopsis, respectively. Therefore, this review mainly focuses on the progress of rice and Arabidopsis. Meanwhile, rice has also been widely demonstrated as one of the most Al-tolerant crop species (Yang et al., 2008; Yamaji et al., 2009; Xia et al., 2010; Yokosho et al., 2011; Huang et al., 2012).

- Phosphate mobility is known to depend greatly on soil redox status and iron speciation. The reduction of iron hydroxide and ferri-phosphate under anaerobic conditions, e.g. in rice culture, will promote the release of phosphate into the soil solution. Any reference to iron speciation and redox state is missing in this paper. Is it not relevant?

Answer: Thanks for your concerns. Yes, phosphate mobility depends greatly on soil redox status and iron speciation. We added the description in the revised version according to your suggestions. Please see the following response, and revised manuscript for the details.

In paddy soils, flooding induces reducing conditions in the rhizosphere region, thereby increasing P availability in acid soils due to the reduction of ferric Pi to more soluble ferrous form, which can cause Pi induced adsorption and precipitation of the metals [130].

- Phosphate is also known to interact with lead (Pb). Precipitation with phosphate is a well-known mechanism for detoxifying lead in both plants and animals. Is there any relationship with the matter discussed in this paper?

Answer: Thanks for your concerns. Yes, Pi can also alleviate Pb toxicity by Pi-Pb precipitation in both plants and animals. in this review, we mainly focus on Pi alleviation of Al, Mn, and Cd toxicities.

Reviewer 3 Report

1. Abstarct section needs some values.

2- Author needs to pay more attention to on novelty statement of your research.

3-Please check the lines 57-59. Acid soils have excessive metals concnetration not deficienct. 

4- ATP-binding cassette stands for what?

5- Its better to add some latest information in support of your highlighted points

6- Its better to add some information in table form. 

Author Response

Responses to the editor and reviewers (Original comments/questions in italics followed by our answers/responses):

Reviewer #3:

  1. Abstarct section needs some values.

Answer: Thanks for your concerns and suggestions. Since it is review paper, we mainly summary the previous literature, but have no data to display.

2- Author needs to pay more attention to on novelty statement of your research.

Answer: Thanks for your concerns and suggestions. Until now, no references comprehensively summary the interactions of P nutrition with all these three toxic metals, particularly possible mechanisms driving P alleviation of these toxicities, along with potential applications for crop improvement on acid soils. This is also the novelty of this review. We added some description on it in Introduction part. Please see the following response, and revised manuscript for the details.

Recent molecular, biochemical, morphological, and physiological reports contribute tremendously to understanding plant P nutrition more clearly and comprehensively. However, mechanisms on P alleviation of toxicities of Al, Mn and the heavy metal Cd, and potential applications for crop improvement has been scarcely explored. This review paper mainly assesses the literature on the role of P in alleviating toxicities of all these three toxic metals. We also review recent advances in the role of P sensing and associated gene regulatory networks to cope with Al, Mn and Cd toxicities. These discussions provide new insights into interactions between P and toxic metals, which can be utilized for crop improvement.

3-Please check the lines 57-59. Acid soils have excessive metals concnetration not deficienct.

Answer: Thanks for your concerns. Acid soils often cooccur with deficiencies in essential nutrient elements, typically P, potassium or calcium, as well as, toxicities of metal elements, such as Al, Mn, Fe and Cd.

4- ATP-binding cassette stands for what?

Answer: Thanks for your concerns. ATP-binding cassette transporter is a membrane transporting protein, which harvests the energy present in cellular ATP to drive the translocation of a structurally diverse set of solutes across cell membranes in bacteria and eukaryotic cells.

5- Its better to add some latest information in support of your highlighted points

Answer: Thanks for your concerns and suggestions. We try to refer to some latest information (Berrios et al., 2019; Li et al., 2019; Wang et al., 2019; Alejandro et al., 2020; Zhao et al., 2020; Park et al., 2021; Noor et al., 2022; Zhao et al., 2022). However, until now, no references comprehensively summary the interactions of P nutrition with all these three toxic metals, particularly possible mechanisms driving P alleviation of these toxicities, along with potential applications for crop improvement on acid soils. This is also the novelty of this review.

6- Its better to add some information in table form.

Answer: Thanks for your concerns and suggestions. We try to employ a schematic model explaining Pi roles in alleviation of Al, Mn and Cd toxicities in soils and plants, and through signaling pathways, which will make the manuscript more clearly and accurately to be understood.

Round 2

Reviewer 1 Report

The author response to one of my comments is:

"Thanks for your concerns and suggestions. We checked the literature and also asked some experts in soil science to try to know the expanse and extent of acid soils particularly cultivated in the world. It is pity that no relevant statistic data is reported except that one original paper published in 1995 by von Uexküll and Mutert in Plant and Soil mentioned that as much as 50% of the world’s potentially arable lands were acidic."

on line 63, I am afraid that the "50% of the world arable lands .........." is quite misleading and should be removed and authors must add to their ms that their review is based on the hydroponic and/or rare find acidic soil studies. The explanation will help readers or practitioners to understand the context and not to generalize/apply it to their neutral or other soil projects.

It would be helpful to see the declaration/statement in the revision before it is accepted for publication.  

Author Response

Responses to the editor and reviewers (Original comments/questions in italics followed by our answers/responses):

Reviewer #1:

"Thanks for your concerns and suggestions. We checked the literature and also asked some experts in soil science to try to know the expanse and extent of acid soils particularly cultivated in the world. It is pity that no relevant statistic data is reported except that one original paper published in 1995 by von Uexküll and Mutert in Plant and Soil mentioned that as much as 50% of the world’s potentially arable lands were acidic."

on line 63, I am afraid that the "50% of the world arable lands .........." is quite misleading and should be removed and authors must add to their ms that their review is based on the hydroponic and/or rare find acidic soil studies. The explanation will help readers or practitioners to understand the context and not to generalize/apply it to their neutral or other soil projects.

Answer: Thank you for your concerns and suggestions. We removed the description about acid soils on Line 63, and also added the description in Conclusion Section to indicate that this review is based on the hydroponic and/or rare find acidic soil studies. Please see the following response, and revised manuscript for the details.

Based on the studies carried out in hydroponic systems and on several acid soils, the role of P in enhancing plant resistance to metal toxicities (Al and heavy metals) has been linked to its immobilization of these toxic elements in the rhizosphere, cell walls and cytosols, as well as, its regulation of organic acid biosynthesis and exudation of organic acid anions, such as citrate and malate.

It would be helpful to see the declaration/statement in the revision before it is accepted for publication. 

Answer: Thank you for your concerns and suggestions. We added the statements at the end of MS. Please see the following response, and Lines 498-501 of the revised manuscript for the details.

Institutional Review Board Statement: Not applicable.

Informed Consent Statement: Not applicable.

Data Availability Statement: No new data were created or analyzed in this study. Data sharing does not apply to this article.